# GC–MS Characterization and Bioactivity Study of *Eucalyptus globulus* Labill. (Myrtaceae) Essential Oils and Their Fractions: Antibacterial and Antioxidant Properties and Molecular Docking Modeling

**DOI:** 10.3390/ph17111552

**Published:** 2024-11-19

**Authors:** Abdessamad Ait benlabchir, Kawtar Fikri-Benbrahim, Amina Moutawalli, Mohammed M. Alanazi, Asma Halmoune, Fatima Zahra Benkhouili, Asmaa Oubihi, Atul Kabra, Elbatoul Hanoune, Hamza Assila, Zineb Benziane Ouaritini

**Affiliations:** 1Laboratory of Natural Substances, Pharmacology, Environment, Modeling, Health and Quality of Life (SNAMOPEQ), Faculty of Sciences, Sidi Mohamed Ben Abdellah University, Fez 30000, Morocco; 2Laboratory of Microbial Biotechnology and Bioactive Molecules, Sciences and Technologies Faculty, Sidi Mohamed Ben Abdellah University, P.O. Box 2202, Imouzzer Road, Fez 30000, Morocco; kawtar.fikribenbrahim@usmba.ac.ma; 3Therapeutic Chemistry Laboratory, Department of Drug Sciences, Faculty of Medicineand Pharmacy, Mohammed V University, Rabat 10100, Morocco; amina_moutawalli@um5.ac.ma (A.M.); fatimazahra_benkhouili@um5.ac.ma (F.Z.B.); 4Department of Pharmaceutical Chemistry, College of Pharmacy, King Saud University, Riyadh 11451, Saudi Arabia; 5Bioactive, Health and Environment Laboratory, Department of Biology, Faculty of Sciences, Moulay Ismail University, Meknes 11201, Morocco; a.halmoune@edu.umi.ac.ma; 6Laboratory of Natural Resources and Sustainable Development Laboratory, Department of Biology, Faculty of Sciences, Ibn Tofail University, Kenitra 14000, Morocco; asmaa.oubihi@uit.ac.ma; 7University Institute of Pharma Sciences, Chandigarh University, Mohali 140413, India; atul.e9963@cumail.in; 8Laboratory of Agricultural Production Improvement, Biotechnology and Environment (LAPABE), Faculty of Sciences, University Mohammed First, Oujda 60000, Morocco; hanoune.elbatoul@ump.ac.ma; 9Laboratory of Medicinal Chemistry, Faculty of Medicine and Pharmacy, Mohammed V University, Rabat 10100, Morocco

**Keywords:** *Eucalyptus globulus*, essential oil, fractions, antioxidant activity, antibacterial properties, molecular docking

## Abstract

Background/Objectives: *Eucalyptus globulus* is a medicinal plant extensively used by the Moroccan population for treating a range of illnesses, especially respiratory conditions. Methods: This study aimed to assess the antioxidant and antibacterial properties of *E. globulus* essential oil and its individual fractions (F1, F2, and F3). Antioxidant activity was evaluated through iron-reducing power, 2,2′-azino-bis (3-ethylbenzothiazoline-6-sulfonic acid) (ABTS), and 2,2-diphenyl-1-picrylhydrazyl (DPPH) assays. Antibacterial activity was tested using disk diffusion and dilution methods, supported by molecular docking studies. Furthermore, GC–MS analysis was conducted on the essential oil and its individual fractions. Results: GC–MS analysis identified the major compounds in the essential oil and its fractions as eucalyptol (62.32–42.60%), globulol (5.9–26.24%), o-cymene (6.89–24.35%), cryptone (7.10–15.95%), terpinen-4-ol (2.43–15.24%), and α-pinene (2.46–7.89%). Fraction F3 displayed the highest antioxidant activity in DPPH (IC_50_ = 3.329 ± 0.054 mg/mL) and ABTS assays (IC_50_ = 3.721 ± 0.027 mg/mL), while fraction F2 was most effective in the FRAP assay (IC_50_ = 1.054 ± 0.008 mg/mL). The essential oil and its fractions also showed antibacterial activity against *Staphylococcus aureus*, *Staphylococcus epidermidis*, *Klebsiella pneumoniae*, *Enterobacter cloacae*, *Escherichia coli*, and *Acinetobacter baumannii*. Molecular docking further corroborated these findings, supporting both antioxidant and antibacterial activities. Conclusions: The present findings demonstrate the antioxidant and antimicrobial properties of *Eucalyptus globulus* essential oil and its fractions, underscoring the need for further research to confirm their medicinal potential and explore pharmaceutical applications.

## 1. Introduction

Eucalyptus, a plant native to Australia, is widely recognized for its diversity, with over 700 species identified. This tree is cultivated for various uses, including wood, gum, and oil, as well as for its aesthetic and therapeutic properties [1]. In Morocco, Eucalyptus is grown in artificial plantations and reforestation projects covering approximately 490,518 hectares, which represents about 5.4% of the country’s total forest area. The total forest area spans around 9,037,714 hectares and includes other introduced species, such as acacias. The cultivation of *Eucalyptus* trees in Morocco began in the Tangier region in the late 19th century, around 1890 [2]. At that time, a local farmer introduced fast-growing Eucalyptus species of Australian origin in the Sidi Yahia region [3].

In recent years, essential oils (EOs) from aromatic and medicinal plants have gained popularity, and their bioactive compounds have found applications in the food, pharmaceutical, medical, and cosmetics industries [4]. The chemical composition of the bioactive components in *Eucalyptus globulus* EO is characterized by a high content of eucalyptol, which ranges from 44% to 84% and is recognized for its antibacterial, antifungal, and antiviral properties. These bioactive properties have made *Eucalyptus globulus* EO a subject of significant pharmacological interest as a complementary or alternative treatment for various infections [5,6]. The limitations of synthetic antioxidants and the growing interest in natural, non-toxic alternatives have led to extensive research into the antioxidant potential of EOs. In addition, EOs contain multiple components, including monoterpenes, sesquiterpenes, alcohols, esters, aldehydes, and ketones, which protect the plant against pests, herbivores, fungi, and bacteria [7].

The antimicrobial efficacy of *Eucalyptus globulus* EO is another focus of our investigation. In light of growing concerns about antibiotic resistance and the limitations of synthetic antimicrobial agents, EOs, like other natural remedies, offer a sustainable and potentially effective solution. *Eucalyptus globulus* oil has demonstrated notable antimicrobial activity against a wide range of bacteria, fungi, and viruses in previous studies. This research builds upon existing knowledge by evaluating the oil’s effectiveness against specific pathogens through methods such as agar diffusion and minimum inhibitory concentration (MIC) assays. These experiments aim to provide valuable insights into the practical applications of *Eucalyptus globulus* EO in pharmaceuticals, personal care products, and disinfectants [8].

Several studies have examined the chemical composition and antibacterial and antioxidant activities of *Eucalyptus globulus* EO extracted using the Clevenger-assisted hydro-distillation technique [9,10,11,12]. The fractionation of essential oils, such as eucalyptus oil, has traditionally been achieved through conventional methods like distillation, solvent extraction, and steam distillation. While these techniques can isolate specific components, they often have significant limitations, particularly in terms of selectivity and preserving heat-sensitive compounds. This study proposes a fractional distillation method using a B-585 glass furnace designed to overcome these limitations by providing precise temperature control and enabling incremental fractionation. However, these essential oils have not previously been fractionated. The present article seeks to provide a comprehensive assessment of the antioxidant and antimicrobial properties of the essential oil (EO) and specific fractions of *Eucalyptus globulus* collected in the Rabat region of Morocco, along with a detailed analysis of their chemical composition. It is worthwhile to mention that this study aims to highlight the therapeutic and preservative potential of these oils through rigorous scientific methodologies, quantifying their effectiveness and identifying the active components responsible for these beneficial effects.

## 2. Results and Discussion

### 2.1. Chemical Composition of Eucalyptus globulus Essential Oil and Its Fractions

The essential oil (EO) of *Eucalyptus globulus* and its fractions were analyzed using gas chromatography–mass spectrometry (GC–MS), resulting in the identification of twenty-nine components that collectively account for approximately 98–99.8% of the total chemical composition (Table 1) (Figure 1). The main constituent of *E. globulus* crude EO is eucalyptol (62.32%), accompanied by other constituents in smaller proportions: p-cymene (8.11%), globulol (5.9%), α-pinene (4.15%), α-fenchol (3.83%), γ-terpinene (3.51%), terpinen-4-ol (2.43%), and β-myrcene (1.51%) (Figure 2), totaling approximately 99.84%. This dominance of eucalyptol has also been reported by Ait-Ouazzou et al. [9] and Vilela et al. [13]. Yet, some authors, such as Hafsa et al. [11], have found that *E. globulus* EO harvested in Tunisia is dominated by p-cymene (18.18%) while eucalyptol represents only 3.16%.

Studies conducted in Portugal by Luís et al. [14] and in Slovakia by Čmiková et al. [15] have found similar eucalyptol contents to our study, with 63.81% and 63.1%, respectively. Conversely, Ait-Ouazzou et al. [9] reported trace amounts of other compounds, such as carvacrol (0.13%) and thymol (0.04%), in the Angad region of Oujda, Morocco, which were not present in our essential oil (EO) composition.

The variations observed in EO chemical composition compared to previous studies can be attributed to several factors, including environmental and climatic conditions, harvesting period, storage conditions, extraction methods, and analysis parameters (e.g., column type, programmed temperature) used to identify EO constituents [16,17]. Chromatographic analysis results indicate that each fraction contains specific major constituents from the crude EO of *E. globulus* (Table 1). Notably, the percentages of p-cymene and terpinen-4-ol increase significantly in fractions F1 and F2, reaching 24.35% and 19.86%, respectively, while the content of eucalyptol decreases to 42.60% and 34.99%, respectively. Although the chemical composition of our EO does not comprise these compounds, they may be present at concentrations below the detection limit of the GC–MS used, as suggested by previous studies on this type of essential oil, such as cryptone (13.10%) and linalool (1.79% and 2.31%), which were present in the fractions and may have resulted from thermal transformation. Fraction 3 (residue) exhibited a distinct chemical profile, characterized by various oxidized substances and a reduced presence of terpene compounds. This progressive fractional distillation protocol was designed to ensure the efficient separation of various constituents while minimizing the thermal degradation of heat-sensitive compounds.

### 2.2. Antioxidant Properties

Since oxidative processes are complex, it is essential to use various assays to verify the antioxidant capacity of the extracts studied. Therefore, we have assessed their free radical scavenging capacity using DPPH, their iron-reducing capacity (FRAP), and their antioxidant capacity with ABTS. Table 2 presents the antioxidant activity results in IC_50_ (50% inhibitory concentration) and EC_50_ (effective concentration that converts 50% of Fe^3+^ to Fe^2+^).

The essential oil (EO) and its fractions demonstrated antioxidant capacity by effectively reducing DPPH and ABTS free radicals, as well as by converting ferric iron to ferrous iron in a dose-dependent manner. Fraction 4 exhibited a DPPH radical-reducing antioxidant capacity with an IC_50_ value of 3329.34 ± 54.68 μg/mL. The antioxidant properties of crude EO, fraction F1, and fraction F2 could not be determined due to non-linearity observed when plotting inhibition percentages against concentration. In comparison to quercetin, the reference standard (IC_50_ = 5.49 ± 0.02 μg/mL), fraction F3 exhibited markedly weak antioxidant activity.

With respect to antioxidant capacity against the ABTS radical, fraction F3 exhibited the highest activity, measuring 3721.91 ± 27.02 μg/mL, while fraction F2 demonstrated the lowest capacity at 9979.31 ± 122.41 μg/mL. The IC_50_ values for crude EO and fraction F1 could not be determined. Compared with ascorbic acid (IC_50_ = 2.52 ± 0.02 μg/mL), the EO and its fractions displayed weak antioxidant activity.

To evaluate the ferric-to-ferrous reducing power of the essential oil (EO) and its fractions, the FRAP test was employed. Fraction F3 exhibited the highest iron-reducing capacity, measuring 1185.48 ± 6.29 μg/mL, followed by fraction F1 at 1054.93 ± 8.95 μg/mL, while the crude EO demonstrated the lowest capacity (EC50 = 22402.20 ± 64.58 μg/mL). In comparison to catechin, the reference standard (EC_50_ = 13.90 ± 0.03 μg/mL), the crude EO and its fractions exhibited weak antioxidant activity.

The activity observed is likely due to the richness of the essential oil (EO) and its fractions in phytocompounds. This activity may result from a synergistic effect among the individual components. For instance, eucalyptol, the predominant compound in both the fractions and the essential oil (EO), has been previously reported by Horvathova et al. to exhibit varying degrees of reducing, radical scavenging, chelating, and DNA-protecting properties [18].

In comparison to literature data, *Eucalyptus globulus* essential oil (EO) and its fractions exhibited greater iron-reducing power, as well as enhanced DPPH and ABTS radical scavenging activities, than the essential oil from *Eucalyptus globulus* sourced in Algeria (FRAP: IC_50_ = 115.39 ± 1.45 mg/mL; DPPH: IC_50_ = 33.33 ± 0.55 mg/mL). However, in the DPPH test, the antioxidant activity of *E. globulus* EO and its fractions was lower than that of commercially available Tunisian EO (IC_50_ = 57 μg/mL) [19] and hydro-distilled EO of Indian *E. citriodora* (IC_50_ = 425.4 ± 6.79 μg/mL) [20]. This activity was higher than that of EO from Turkey (IC_50_ > 10 mg/mL) [21] and *E. globulus* EO from Algeria (IC_50_ = 33.33 ± 0.55 mg/mL) [12], but lower than that of aqueous extracts of *E. globulus* leaves from Portugal (IC_50_ = 426.8 μg/mL) [22]. Additionally, the activity was superior to that of the essential oil (EO) from Algerian *Eucalyptus globulus* fruits (IC_50_ = 27.0 ± 0.2 μg/mL) [8].

When evaluated using the FRAP test, the antioxidant power of *Eucalyptus globulus* essential oil (EO) and its fractions was lower than that of Tunisian EO (IC_50_ = 48 μg/mL) [19] and Indian *E. citriodora* EO (IC_50_ = 87.3 ± 9.27 μg/mL) [20], but higher than that of Algerian EO (IC50 = 115.39 ± 1.45 mg/mL) [12]. Furthermore, it exceeded the antioxidant power of *E. globulus* fruits from Algeria (IC_50_ = 32.8 ± 1.8 mg/mL) [8].

It is noted that this study found that *E. globulus* essential oil (EO) from Morocco exhibited variations in antioxidant efficacy compared to EOs of the same or other species from different countries. This variation can be attributed to differences in chemical composition, environmental and agronomic factors, plant age, geoclimatic conditions, and including extraction and storage conditions [23].

### 2.3. Antibacterial Activity

According to Table 3 and Figure 3, statistical analysis shows that for *Staphylococcus aureus*, the CEO, and F1 treatments showed no significant difference, but CEO differed significantly from F2 and F3, as did F1 compared with F2 and F3, possibly indicating greater efficacy of F2 and F3 compared with crude EO and F1. On the other hand, for *Staphylococcus epidermidis*, CEO was significantly different from F1 and F2, but not from F3. The differences between F1 and F2 and between F1 and F3 were also significant, suggesting that F1 may be less effective against this bacterium, marking a notable distinction between treatments for these two *Staphylococcus* species.

For *Klebsiella pneumoniae*, CEO was significantly different from F1, F2, and F3, while there was no significant difference between F1, F2, and F3 themselves, indicating that these three treatments could be of similar efficacy. For *Escherichia coli*, CEO did not differ significantly from F1 and F2, but it differed significantly from F3. Moreover, the F1 vs. F3 and F2 vs. F3 comparisons also showed differences, suggesting a possible superiority of F3 for this bacterium.

For *Acinetobacter baumannii*, no significant difference was observed between CEO, F1, and F3, but a difference exist between CEO and F2, as well as between certain treatments, notably between F2 and F3, which could indicate a better efficacy of F2 for this bacterium. Finally, for *Enterobacter cloacae*, no significant difference was observed between treatments, suggesting comparable efficacy of all treatments evaluated against this bacterium.

The antibacterial activity of the essential oil (EO) and its fractions extracted from *Eucalyptus globulus* leaves was evaluated by measuring the diameters of inhibition zones, which indicate areas where bacterial growth was prevented by the EO. Overall, significant inhibitory effects of the EO were observed against both Gram-positive and Gram-negative bacteria, with inhibition zone diameters ranging from 8 to 46 mm, as shown in Table 3. This variation depends on the concentration of the active compounds and the bacterial strains tested.

Table 3 presents the antibacterial activity of the essential oil (EO) and its fractions derived from the leaves of *Eucalyptus globulus* against various bacterial strains. The EO exhibited growth inhibition zones, measured in millimeters, against strains such as *Staphylococcus aureus*, *Staphylococcus epidermidis*, *Klebsiella pneumoniae, Enterobacter cloacae*, *Escherichia coli*, and *Acinetobacter baumannii.* Fractions F2 and F3 demonstrated significant inhibitory power against the *Staphylococcus aureus* strain, with inhibition zones measuring 46 mm and 45 mm, respectively. In comparison, the reference antibiotics ampicillin and gentamicin exhibited inhibition zones of 7 mm and 22 mm, respectively. Crude EO (CEO) and fraction F1 showed inhibition zones of 14 mm and 13 mm against *Staphylococcus aureus*, as well as zones of 11 mm and 13 mm against *Staphylococcus epidermidis*, while ampicillin displayed a moderate zone of 7 mm and proved inactive against *Staphylococcus epidermidis*. The results indicated that CEO and its three fractions exhibited moderate inhibition zones against Gram-positive strains, ranging from 8 mm to 11 mm against *Klebsiella pneumoniae* and *Escherichia coli*. In contrast, the reference antibiotic gentamicin displayed inhibition zones of 19.5 mm and 20 mm, respectively, while ampicillin and penicillin were ineffective against Gram-positive bacteria.

In response to the alarming rise in antibiotic resistance, the prudent use of plants and their derivatives, such as essential oils (EOs), has emerged as a promising strategy. This approach provides targeted and effective mechanisms of action against resistant pathogens, reflecting the immense potential of phytotherapy in the post-antibiotic era. *Eucalyptus globulus* essential oils (EOs) and their fractions demonstrate significant zones of inhibition against the specific bacteria tested when analyzed using the agar diffusion method. According to Table 3, fractions F2 and F3 exhibited inhibition zones of up to 46 mm and 45 mm against *Staphylococcus aureus*, surpassing the effectiveness of the reference antibiotics ampicillin and gentamicin, which showed values of 7 mm and 22 mm, respectively. In contrast, the complete essential oil (EO) CEO only produced an inhibition zone of 14 mm. Several scientific studies have focused on the antibacterial efficacy of *Eucalyptus globulus* essential oils against various bacterial strains. For instance, a study conducted by Shah demonstrated that the essential oil (EO) exhibited an inhibition zone of 19 mm against both *Staphylococcus epidermidis* and *Staphylococcus aureus* strains [24]. In comparison, against the *Staphylococcus aureus* strain, the EO induced an inhibition zone of 5.67 ± 0.58 mm according to [15], and a zone of 20.4 ± 0.2 mm as documented by [25]. The CEO and its F1 fraction have also demonstrated significant inhibition zones against *Staphylococcus epidermidis* and *Acinetobacter baumannii*, with values of 11 mm and 13 mm, and 13 mm and 12.5 mm, respectively. To date, research has primarily focused on the antimicrobial activity of *Eucalyptus globulus* essential oil as a whole, overlooking the specific effects of its various fractions. Our recent study addresses this gap by demonstrating that certain isolated fractions of *Eucalyptus globulus* essential oil (EO) possess greater antimicrobial activity than the complete oil, thus offering new avenues for optimizing natural antibacterial treatments.

Table 4 presents the minimum bactericidal concentrations (MBCs) of essential oils (EOs) extracted from Eucalyptus leaves. The results are expressed in micrograms per milliliter (µg/mL). It is observed that fractions F2 and F3 exhibit the lowest MBC values against the *Staphylococcus aureus* strain, with a concentration of 2 µg/mL, suggesting strong antimicrobial activity. However, the C and F3 oils show moderate activity, recording MBC values of 10 µg/mL against certain tested strains.

The determination of minimum inhibitory concentration (MIC) and minimum bactericidal concentration (MBC) is crucial for evaluating the effectiveness of antimicrobial agents. These parameters not only quantify antimicrobial activity but also differentiate between bacteriostatic and bactericidal effects, which is essential for understanding the agents’ mechanisms of action. The results obtained from these determinations are vital for making relevant comparisons with existing literature.

In our present study, we have observed that fractions F1, F2, and F3 demonstrated notable minimum inhibitory concentrations (MICs) of approximately 2, 2, and 1 µg/mL, respectively, against the *Staphylococcus aureus* strain. Furthermore, F3 showed significant potency, recording an MIC of approximately 4 µg/mL against *Staphylococcus epidermidis* and *Acinetobacter baumannii* strains. The complete essential oil (EO) of *Eucalyptus globulus* (CEO) exhibited an MIC of approximately 10 µg/mL against both *Staphylococcus aureus* and *Staphylococcus epidermidis*, and greater than 10 µg/mL against Gram-negative strains. Our findings indicate that the essential oils (EOs) extracted from *Eucalyptus globulus* leaves demonstrated significant antimicrobial activity, as evidenced by the determination of minimum inhibitory concentration (MIC) values. MIC is defined as the minimum concentration of essential oil required to inhibit the growth of the studied organism. Table 4 presents the MIC results from our study, with values ranging from 1 to 10 µg/mL. These values highlight the antimicrobial potency of the essential oil (EO) and its fractions, demonstrating significant inhibitory effects against various bacterial strains, particularly *Staphylococcus aureus* and *Staphylococcus epidermidis*. These findings underscore the potential of *Eucalyptus globulus* EO as a natural antimicrobial agent and reveal the efficacy of specific EO fractions in achieving targeted antibacterial activity.

The analysis demonstrated that the F3 fraction of *Eucalyptus globulus* essential oil exhibited significant antibacterial properties. Specifically, it recorded a minimum inhibitory concentration (MIC) of only 1 µg/mL against *Staphylococcus aureus*, indicating high efficacy. However, the same fraction displayed a MIC of 4 µg/mL against *Staphylococcus epidermidis*. Regarding Gram-negative bacteria, most of the tested essential oils did not show effectiveness against *Klebsiella pneumoniae*, *Enterobacter cloacae*, *Escherichia coli*, and *Acinetobacter baumannii*, with MIC values equal to or greater than 10 µg/mL. Nevertheless, the F3 fraction distinguished itself by demonstrating a MIC of 4 µg/mL against *Acinetobacter baumannii*, highlighting its specific antimicrobial potential.

Unlike the minimum inhibitory concentration (MIC), which inhibits bacterial growth, the minimum bactericidal concentration (MBC) is a more stringent measure indicating the bactericidal effectiveness of an antimicrobial substance. In our study, fractions F2 and F3 demonstrated an MBC of 2 µg/mL, a concentration sufficient to eliminate the targeted bacteria. According to Assaggaf et al., the crude essential oil recorded MIC and MBC values of approximately 1 and 0.5 µg/mL against *Escherichia coli* and *Staphylococcus aureus*, respectively [26]. Our recorded MIC results were 2.04 and 2.07 µg/mL, while the MBC values were approximately 4.36 and 4.26 µg/mL against *Staphylococcus epidermidis* and *Staphylococcus aureus*, respectively [24]. Other studies have indicated that the MIC of the essential oil varies between 0.5 and 8 µg/mL against *S. aureus* and *E. coli*, with an MBC around 16 µg/mL for both strains [25]. The antibacterial efficacy of essential oils (EOs) is primarily associated with their chemical composition and the specific strains of microorganisms targeted. Much of the antibacterial activity in EOs is attributed to oxygenated monoterpenes, although certain hydrocarbon compounds can also exhibit antibacterial properties [27].

The antimicrobial activity of the two fractions, F2 and F3, from *E. globulus* is primarily attributed to their richness in bioactive compounds. According to Table 4, both fractions exhibit the presence of eucalyptol. This main constituent, eucalyptol (1,8-cineole), is recognized for its antibacterial action. This molecule disrupts the structure and function of the cell membranes of microorganisms, leading to altered permeability and potentially lysis. Additionally, eucalyptol inhibits protein and nucleic acid synthesis, preventing the multiplication of bacteria and viruses. Besides eucalyptol, other components such as α-pinene, o-cymene, linalool, and globulol also contribute to the antimicrobial efficacy of the oil, either individually or synergistically. This synergy among different components offers a promising alternative to traditional antibiotics, particularly in the context of increasing antimicrobial drug resistance.

### 2.4. Molecular Docking

#### 2.4.1. Human Peroxiredoxin 5 Enzyme PRDX5 (PDB ID: 1H2D)

The molecular docking results (Table 5 and Figure 4) indicate that all tested volatile oil components exhibit strong affinity for the PRDX5 enzyme, with binding energies ranging from −4.2 to −4.4 kcal/mol. Among the components, cryptone, o-cymene, and terpinen-4-ol demonstrated the highest binding affinities. These docking scores can be attributed to the diverse molecular interactions observed between the volatile oil components and the amino acids at the active site of PRDX5.

For instance, o-cymene exhibits multiple types of interactions with the amino acids of PRDX5, including three hydrophobic interactions, alkyl interactions, and π–alkyl interactions between the methyl groups and the following residues: PRO 40 (5.13 Å), PRO 45 (4.50 Å), LEU 116 (4.11 Å), ILE 119 (5.16 Å, 5.13 Å), and PHE 120 (4.61 Å, 5.29 Å). Additionally, a π–π T-shaped interaction is observed between the o-cymene benzene ring and PHE 120 (4.87 Å). The benzene ring of o-cymene also engages in a π–sulfur interaction with CYS 47 (5.92 Å). Notably, interactions with the same amino acids were common across all volatile oil components, which may explain the similar binding energies observed.

#### 2.4.2. For the DNA Gyrase from *E. coli* (PDB ID: 3G7B)

The docking results (Table 5, Figure 5) reveal that p-cymen-2-ol exhibited the highest binding energy at −5.3 kcal/mol, likely due to its diverse molecular interactions with 3G7B, including three alkyl-type interactions with residues ARG 76 (4.41 Å), PRO 76 (4.92 Å), and ILE 78 (4.58 Å). o-Cymene, with a binding energy of −5.2 kcal/mol, engaged in multiple hydrophobic interactions with ILE 78 (4.22 Å, 4.76 Å, 5.38 Å) and ILE 94 (4.58 Å), as well as a π–anion interaction between its benzene ring and GLU 50 (4.18 Å).

Terpinen-4-ol and cryptone had affinity energies of −5.1 kcal/mol and −5.0 kcal/mol, respectively, with varied interactions: Terpinen-4-ol formed alkyl interactions with ILE 78 (5.40, 4.73 Å) and PRO 79 (4.72 Å), plus a hydrogen bond with GLU 50 (2.85 Å); cryptone was stabilized by extensive alkyl interactions with PRO 79 (4.69 Å), ILE 78 (3.92, 4.17, 4.58 Å), and ILE 94 (4.47, 4.71 Å), and two hydrogen bonds with GLY 77 (2.71 Å) and THR 165 (2.334 Å). The final two compounds, cucalyptol and α-pinene, displayed slightly lower binding energies of −4.6 and −4.7 kcal/mol, respectively, both interacting with ILE 78.

This range of binding energies and interactions underscores the variable affinities and stabilization mechanisms of each compound within the binding pocket.

#### 2.4.3. For the DNA Gyrase from *S. aureus* (PDB ID: 6F86)

The highest binding energy recorded for the volatile oil components with *Staphylococcus aureus* (6F86) (Table 5, Figure 6) was observed for p-cymen-2-ol at −5.8 kcal/mol, followed by terpinen-4-ol at −5.6 kcal/mol, cryptone at −5.4 kcal/mol, o-cymene at −5.2 kcal/mol, eucalyptol at −4.7 kcal/mol, and finally α-pinene at −4.6 kcal/mol.

In the docking studies, p-cymen-2-ol demonstrated two alkyl interactions with ILE 86 (4.56, 4.73 Å) and ILE 175 (4.12, 4.35 Å), alongside a conventional hydrogen bond between its alcohol group and SER 55 (2.97 Å). Terpinen-4-ol presented three hydrophobic interactions with ILE 51 (4.19, 4.98 Å), ILE 86 (4.85 Å), and ILE 175 (4.36, 4.47 Å), as well as a hydrogen bond with ASP 81 (2.86 Å). Cryptone displayed only alkyl interactions with ILE 86 (4.81 Å) and ILE 175 (4.28, 4.59, 4.51, 5.01 Å), while o-cymene interacted solely with ILE 86, forming two alkyl interactions with its methyl groups (3.97, 5.93 Å) and a π–σ interaction with its benzene ring (3.74 Å). Eucalyptol and α-pinene both exhibited alkyl interactions with ILE 86, with eucalyptol also forming an additional alkyl interaction with ILE 102 (4.18 Å). This overview highlights each compound’s unique binding interactions and provides a basis for comparing their affinity and specificity for the target residues.

#### 2.4.4. For the OmpA-like Protein of *A. Baumannii* (PDB ID: 3TD3)

The docking results (Table 5, Figure 7) showed that all the volatile oil components presented an interesting affinity result toward (3TD3). Eucalyptol exhibited the highest binding energy of −4.8 kcal/mol. This docking score is attributed to the hydrophobe–alkyl type of interaction between eucalyptol and 3TD3 amino acids, LEU 278 (3.92 Å, 4.49 Å, 5.02 Å), and LEU 282 (4.39 Å, 4.98 Å, 5.10 Å). α-Pinene and terpinen-4-ol, on the other hand, exhibited the second highest affinity energy of approximately −4.7 kcal/mol and engaged in several hydrophobic interactions with the residues LEU 278, ARG 281, LEU 282, and ALA 285. Cryptone demonstrated an affinity energy of the order −4.5 kcal/mol; this energy is attributed to the alkyl type of interaction with the residues LEU 278 (4.06 Å), ARG 281 (4.41 Å), LEU 282 (4.23 Å, 4.69 Å), and ALA 285 (4.16 Å). Finally, o-cymene and p-cymen-2-ol had the same binding energy toward (3TD3) of an order of −4.4 kcal/mol. The interaction between o-cymene and (3TD3) amino acids was stabilized by the alkyl and π–alkyl type of interactions with LEU 278 (3.81 Å, 4.13 Å), LEU 282 (3.81 Å, 4.58 Å), ALA 285 (4.71 Å), and amide–π interaction with ARG 281 (4.60 Å). p-Cymen-2-ol exhibited different hydrophobic interactions with LEU 278, ARG 281, LEU 282, ALA 285, ALA 302, and one hydrogen bond with ASN 237 (2.04 Å).

The in silico study showed very potent results that align with our in vitro results for antioxidant and antibacterial activities.

## 3. Materials and Methods

### 3.1. Chemicals and Reagents

Hexane, methanol, DPPH, quercetin, potassium persulfate, ABTS radical, sodium phosphate buffer, potassium ferricyanide, trichloroacetic acid, ferric chloride (FeCl_3_), Catechin, whatman filter paper, penicillin, gentamicin, ampicillin (The chemicals used in this study were purchased from Biotechnics Solution, based in Casablanca, Morocco).

### 3.2. Plant Material and Extraction of the Essential Oil

The plant material consisted of *Eucalyptus globulus* leaves collected from the Rabat region of Morocco in March 2023.

The plant material was identified by Mohamed El Kadiri, a botanist at the Faculty of Sciences in Tétouan, Abdelmalek Essaâdi University, Morocco. An herbarium specimen has been deposited with the reference number 1-130.881.

A total of 200 g of freshly harvested leaves was subjected to hydro-distillation for 3 h using a Clevenger apparatus (Biotechnics Solution, Casablanca, Morocco). The essential oil (EO) obtained was then stored at 4 °C for later use in experiments.

### 3.3. Fractionation of Essential Oil

The fractional distillation process was conducted using a B-585 glass furnace (BUCHI Flawil, Suisse). All glass balls were placed in the furnace, and a quantity of 10 mL of essential oil was added to one of the glass balls. The mixture was then boiled at temperatures of 160 °C for fraction 1 (F1) and 180 °C for fraction 2 (F2), each for 10 min. The residual fraction 3 (F3) was collected from the initial ball after the fractionation process.

### 3.4. GC–MS Analysis

The chemical composition of the essential oils (EOs) was determined, by GC-MS, using a Varian capillary column (PerkinElmer, Waltham, MA, USA) with the following specifications: a length of 50 m, a diameter of 0.32 mm, and a film thickness of 1.25 µm (TR5-CPSIL-5CB). The column temperature was increased from 40 °C to 280 °C at a rate of 5 °C per minute. The injector temperature was maintained at 260 °C, and the flame ionization detector (FID) (PerkinElmer, Waltham, MA, USA) was also set to 260 °C. Helium was used as the carrier gas, flowing at a rate of 1 mL/min. A volume of 1 microliter of the oil, diluted in hexane to a concentration of 10%, was injected into the column. The chemical components of the oils were identified by comparing them with substances obtained from the Adams table [28] and the NIST MS Library.

### 3.5. Determination of Antioxidant Activity

#### 3.5.1. Free Radical Scavenging Activity of DPPH

Following the protocol developed by Sahin et al. [29], the 2,2-diphenyl-1-picrylhydrazyl (DPPH) assay was employed to evaluate the antioxidant activity of *Eucalyptus globulus* essential oil (EO) and its investigated fractions. This procedure involved combining 50 μL of each EO and its respective fractions at different concentrations with 2 mL of a DPPH solution at a concentration of 0.0023% (60 μM DPPH in methanol). After rapid stirring, the mixture was allowed to stand for 20 min at room temperature, protected from light. The absorbance of the resulting solution was measured at 517 nm. A control sample (DPPH without extracts) was also prepared. Quercetin was used as the standard antioxidant, with concentrations ranging from 0.38 to 6.09 µg/mL. The percentage of inhibition was calculated using the following formula:$$\%\;Inhibition\;of\;DPPH\;activity=\frac{Absorbance\;of\;control-Absorbance\;of\;sample}{Absorbance\;of\;control}\times 100$$

#### 3.5.2. ABTS Free Radical Scavenging Activity

The ABTS radical scavenging assay was conducted according to the procedure described in ref. [30]. The ABTS radical was generated by oxidizing ABTS with potassium persulfate. Briefly, an ABTS solution (7 mM) was mixed with a potassium persulfate solution (70 mM) in equal volumes to produce ABTS cations. This mixture was stored in the absence of light at ambient temperature for 16 h. Prior to analysis, the ABTS cation radical was diluted with methanol to achieve a starting absorbance of approximately 0.700 at a wavelength of 734 nm. In this experiment, varying amounts of each fraction and the crude essential oil (100 μL) were introduced into a 2 mL solution of ABTS. The absorbance was measured at 734 nm, and the percentage of inhibition was calculated using the method described for the DPPH assay.

#### 3.5.3. Ferric Reducing Antioxidant Power Test (FRAP)

The reducing activity of each fraction and their crude essential oil (EO) was measured spectrophotometrically using the Oyaizu method [31]. Different concentrations of each fraction and their crude EO (0.2 mL) were mixed with 2.5 mL of sodium phosphate buffer (pH = 6.6) and 2.5 mL of potassium ferricyanide (1% *w/v*) (K_3_Fe(CN)_6_). After a 20 min incubation at 50 °C, 2.5 mL of 10% (*w/v*) trichloroacetic acid was added to the solution. Approximately 2.5 mL of this mixture was combined with 2.5 mL of distilled water and 0.5 mL of ferric chloride (FeCl_3_) at 0.1% (*w/v*) iron content. The intensity of the resulting blue-green hue was measured at 700 nm. Catechin (0.65 to 21.39 μg/mL) was used as a control.

### 3.6. In Vitro Antimicrobial Activity

#### 3.6.1. Microorganisms Studied

The germs selected to evaluate the antimicrobial activity of essential oils are chosen based on their pathogenicity and resistance to antibiotics. Specifically, there are two Gram-positive bacteria (*Staphylococcus aureus* and *Staphylococcus epidermidis*) and four Gram-negative bacteria (*Escherichia coli*, *Acinetobacter baumannii*, *Klebsiella pneumoniae*, and *Enterobacter cloacae*). These strains are provided by the microbiology department of the medical analysis laboratory.

#### 3.6.2. Disk Diffusion Method

The antibacterial activity was assessed using the disk diffusion method on Mueller–Hinton agar plates [32]. A microbial suspension with an optical density of 1 McFarland was evenly spread over the surface of the agar medium in a Petri dish. Whatman filter paper disks, 6 mm in diameter and previously sterilized by autoclaving, were impregnated with the essential oil (EO) being tested. These prepared disks were then placed onto the surface of the agar medium, and the entire setup was incubated at 37 °C for 24 h. Upon application, the essential oil diffuses uniformly from the impregnated disks. After incubation, the presence of a circular zone of inhibition around the disks, where no microbial growth occurs, indicates the susceptibility of the microorganisms to the oil. The larger the zone of inhibition, the more sensitive the microorganism is to the tested oil. Penicillin (5 μg), gentamicin (500 μg), and ampicillin (10 μg) were used individually as positive controls.

#### 3.6.3. Minimum Inhibitory Concentration (MIC) and Minimum Bactericidal Concentration (MBC)

The minimum inhibitory concentration (MIC) and minimum bactericidal concentration (MBC) were determined using the solid macro-dilution technique [33]. The essential oil (EO) and its fractions were emulsified with a 0.2% agar solution to achieve homogeneous distribution within the medium. Dilutions were prepared in this agar solution at concentrations of 100 µL/mL, 40 µL/mL, 20 µL/mL, 10 µL/mL, 5 µL/mL, 3.3 µL/mL, and 2 µL/mL. In test tubes, each containing 13.5 mL of solid Mueller–Hinton medium, 1.5 mL of each dilution was aseptically added to achieve final concentrations of 10 µL/mL, 4 µL/mL, 2 µL/mL, 1 µL/mL, 0.5 µL/mL, 0.33 µL/mL, and 0.2 µL/mL. The contents of each tube were immediately poured into sterile Petri dishes after 15 s of agitation. Control samples containing only the culture medium were included. Inoculation was performed by streaking with a calibrated platinum loop to ensure the same volume of inoculum was collected. Incubation was carried out at 37 °C for 24 h.

The MIC was determined based on the first dish in the series that was devoid of bacterial growth. Each experiment was repeated three times.

The minimum bactericidal concentration (MBC) is determined the day after reading the minimum inhibitory concentration (MIC). To do this, a streak inoculation is performed on Mueller–Hinton agar using samples taken from each dish where no growth is visible to the naked eye. The inoculated dishes are then incubated for 24 h at 37 °C for bacteria and for 48 h at 30 °C for yeasts. The MBC corresponds to the lowest concentration at which no subculture is observed [32]. The MBC (%, *v/v*) of the essential oil is determined from the first dish that shows no bacterial growth.

### 3.7. Molecular Docking Study

To validate the experimental results and gain a deeper understanding of the potential interactions underlying the antioxidant and antibacterial properties, specific volatile oil components were examined through molecular docking into the active sites of human peroxiredoxin 5 enzyme (PRDX5, PDB ID: 1H2D), *E. coli* DNA gyrase (PDB ID: 3G7B), *Staphylococcus aureus* DNA gyrase (PDB ID: 6F86), and OmpA-like domain from *Acinetobacter baumannii* (PDB ID: 3TD3) [34,35,36] downloaded from the PDB database.

The 3D structures of eucalyptol, α-pinene, cryptone, o-cymene, p-cymen-2-ol, and terpinen-4-ol were downloaded from the PubChem database.

The preparation of the proteins (1H2D, 3G7B, 6F86) was performed following our previous studies [37], using the AutoDock Toolkit (ADT) [38]. This preparation involved deleting water molecules, adding non-polar hydrogen atoms, filling in any missing atoms on the residues, distributing the Kollman charge evenly across all residues, and keeping the remaining docking parameters at their default settings.

Potential binding sites of the human peroxiredoxin 5 enzyme were identified using Discovery Studio software (v16.1.0.15350) [39], based on the receptor structure without the prior introduction of the ligand. Before running the docking, we prepared the grid box using AutoGrid (California, United States), placing it on the enzyme according to its XYZ coordinates, with dimensions of (40, 40, 40) Å and a default grid spacing of 0.375 Å.

For the validation of molecular docking, we employed the redocking method. The active site was prepared by removing complex ligands from the receptor structures and subsequently redocking them into the protein’s active site. The root mean square deviation (RMSD) between the redocked conformation and the original crystallographic conformation of the compound was found to be approximately 1.0 Å, confirming the reliability of the docking method in replicating the experimentally observed binding mode for the receptor [40].

### 3.8. Statistical Analysis

Statistical analysis was conducted using GraphPad Prism v8 software. The data presented are mean values ± standard deviation from three independent extractions. The data were subjected to a one-way analysis of variance (ANOVA), and mean values were compared using Tukey’s test at a significance level of *p* < 0.05.

## 4. Conclusions

The presence of essential oils in the pharmaceutical and food industries calls for a thorough examination of their composition and characteristics, including their antioxidant potential to reduce lipid oxidation and prevent rancidity, as well as their antibacterial efficacy in inhibiting microbial growth. This study focused on evaluating the antioxidant capacity of EOs to inhibit lipid oxidation and prevent rancidity, as well as their antibacterial efficacy against various microorganisms. We investigated the chemical composition, fractionation, antioxidant activity, and antimicrobial properties of *Eucalyptus globulus* essential oils, extracted via hydro-distillation. A total of twenty-nine constituents were identified in the crude oil and its fractions. The fractional distillation process produced three distinct fractions, with the third fraction exhibiting significantly enhanced antioxidant activity against DPPH and ABTS compared to the crude EOs. Additionally, this fraction demonstrated notable antibacterial activity against both pathogenic and spoilage microorganisms. Our findings revealed that all tested strains were sensitive to *Eucalyptus globulus* essential oils, particularly Gram-negative bacteria, which are often resistant to conventional antibiotics. These results underscore the antioxidant and antimicrobial properties of *Eucalyptus globulus* EOs and their fractions, highlighting the need for further studies to validate their medicinal potential and investigate their applications in pharmaceuticals.

## Figures and Tables

**Figure 1 pharmaceuticals-17-01552-f001:**
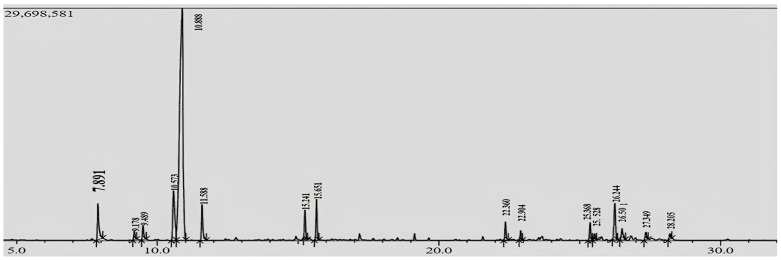
GC–MS chromatogram of *Eucalyptus globulus* EO and its fractions.

**Figure 2 pharmaceuticals-17-01552-f002:**
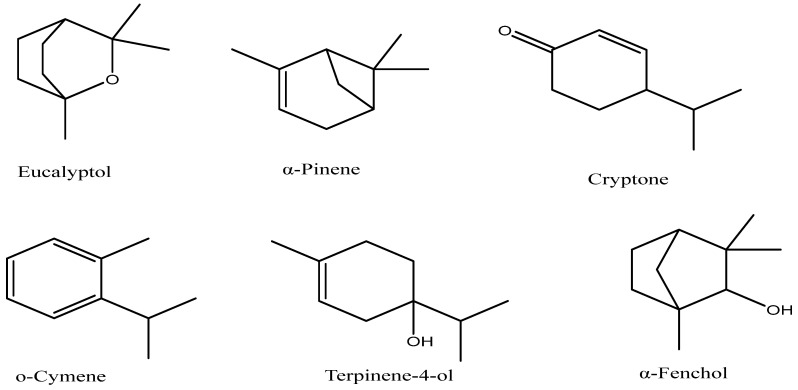
Predominant volatile components in *E. globulus* essential oil and its fractions.

**Figure 3 pharmaceuticals-17-01552-f003:**
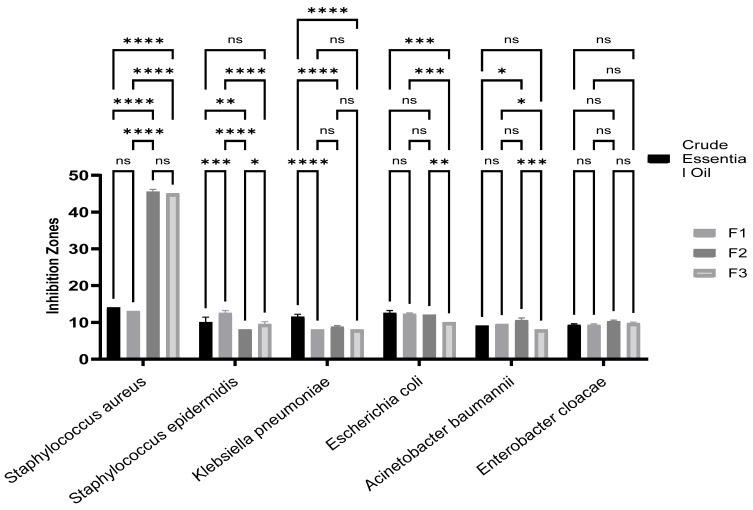
Comparative analysis of inhibition zones of essential oil and its fractions (F1, F2, F3) against various strains of microorganisms; “****” for *p* < 0.0001 (highly significant),“ ***” for *p* < 0.001,“ **” for *p* < 0.01, “ *” for *p* < 0.05,“ ns” for not significant.

**Figure 4 pharmaceuticals-17-01552-f004:**
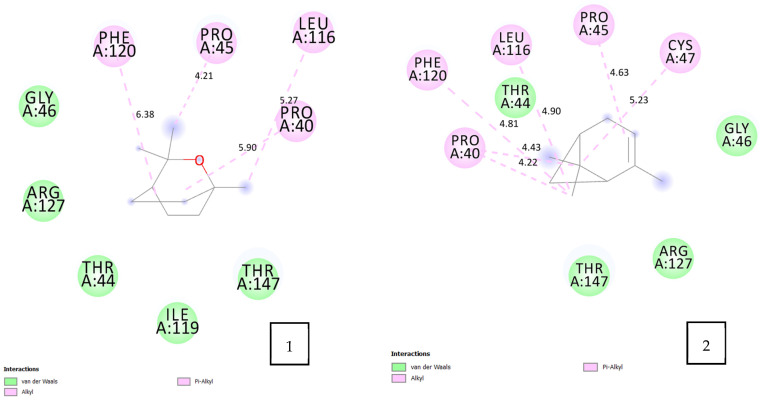
2D diagram of the interaction between the human peroxiredoxin 5 receptor (PRDX5, PDB ID: 1H2D) and the selected volatile oil compounds. 1: eucalyptol; 2: α-pinene; 3: cryptone; 4: o-cymene; 5: p-cymen-2-ol; 6: terpinen-4-ol.

**Figure 5 pharmaceuticals-17-01552-f005:**
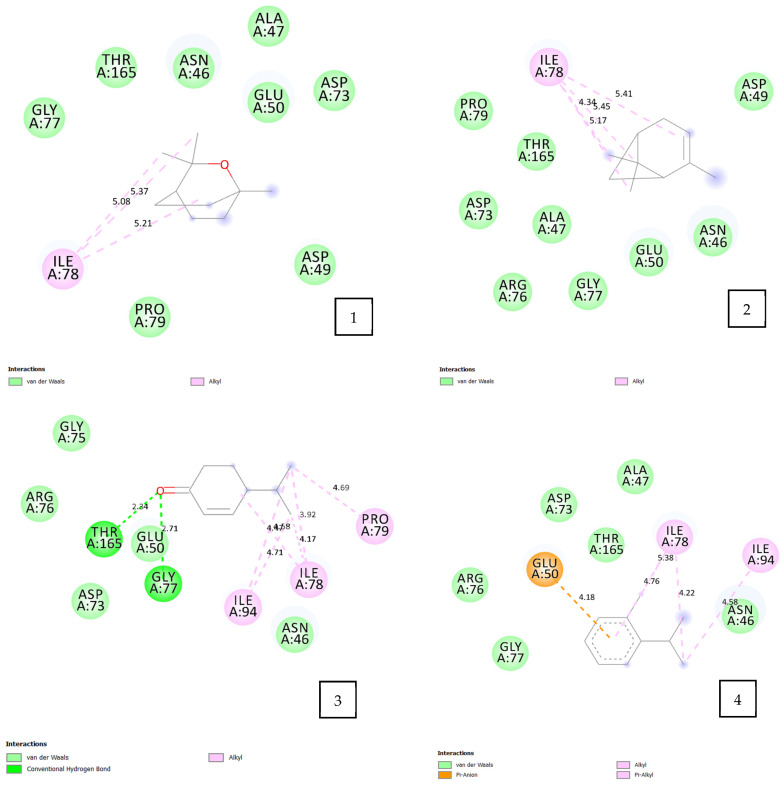
2D diagram of the interaction of DNA gyrase from *E. coli* with the selected volatile oil compounds. 1: eucalyptol; 2: α-pinene; 3: cryptone; 4: o-cymene; 5: p-cymen-2-ol; 6: terpinen-4-ol.

**Figure 6 pharmaceuticals-17-01552-f006:**
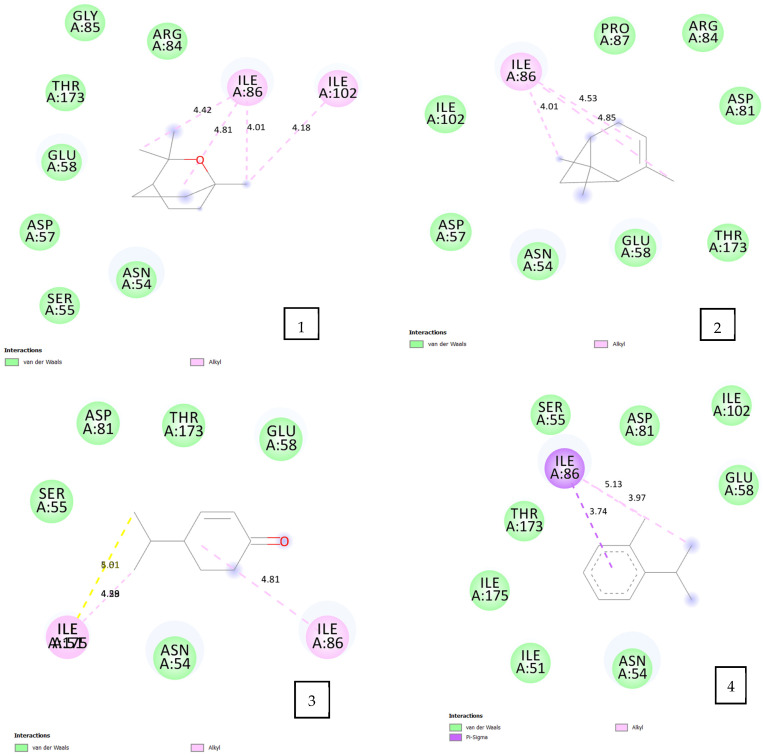
2D diagram of the interaction of DNA gyrase from *Staphylococcus aureus* with the selected volatile oil compounds. The compounds include the following: 1: eucalyptol; 2: α-pinene; 3: cryptone; 4: o-cymene; 5: p-cymen-2-ol; 6: terpinen-4-ol.

**Figure 7 pharmaceuticals-17-01552-f007:**
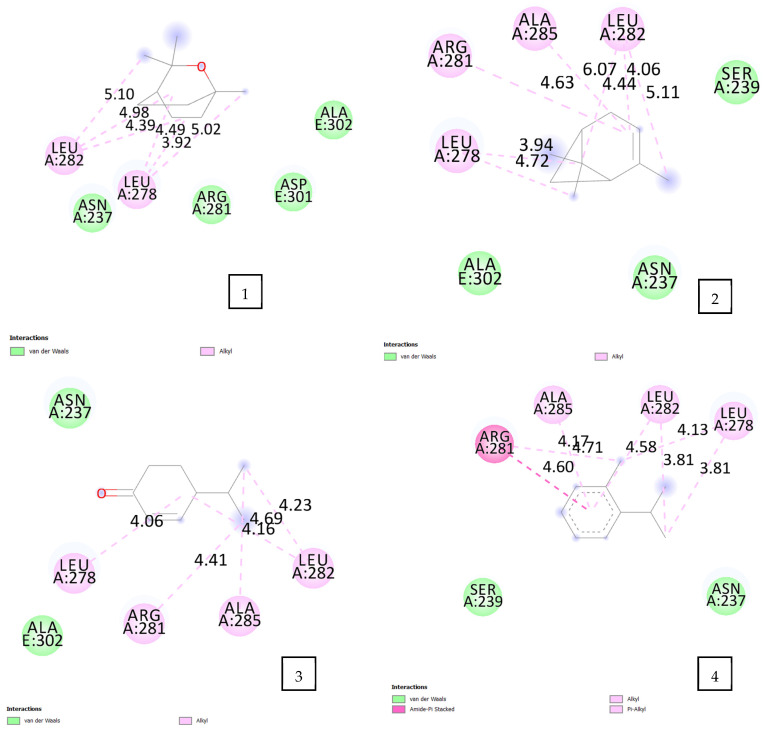
2D diagram of the interaction of the OmpA-like protein of *Acinetobacter baumannii* with the selected volatile oil compounds. 1: eucalyptol; 2: α-pinene; 3: cryptone; 4: o-cymene; 5: p-cymen-2-ol; 6: terpinen-4-ol.

**Table 1 pharmaceuticals-17-01552-t001:** Chemical composition of *Eucalyptus globulus* EO and its fractions analyzed by GC–MS.

N.pic	Compounds	RT (min)	* RI Retention Indices	RI Retention Index Libraries	Similarity Index (%)	Relative Percentage of *Eucalyptus globulus* and Its Fractions			
						CEO	(F1)	(F2)	(F3)
1	α-Thujene	7.659	902	938	96	-	1.30	-	-
2	α-Pinene	7.891	948	939	97	4.15	2.46	-	-
3	β-Pinene	9.178	943	981	96	0.86	-	-	-
4	Octamethylcyclotetrasiloxane	9.290	827	-	93	-	-	-	2.97
5	β-Myrcene	9.489	958	992	95	1.51	-	-	-
6	p-Cymene	10.573	1042	1027	95	8.11	24.35	19.86	6.89
7	Limonene	10.700	1018	1033	92	-	2.96	-	-
8	Eucalyptol	10.888	1059	1030	94	62.32	42.60	34.99	10.65
9	γ-Terpinene	11.588	998	1074	96	3.51	-	-	-
10	Linalool	12.784	1082	1100	95	-	1.79	2.31	-
11	p-Menth-2-en-1-ol	13.557	1109	1108	93	-	-	0.97	-
12	Decamethylcyclopentasiloxane	13.910	-	-	-	-	-	-	31.70
13	Terpinen-4-ol	15.241	1137	1179	94	2.43	4.82	8.47	-
14	Cryptone	15.388	1069	1156	93	-	13.10	15.95	7.10
15	α-Fenchol	15.651	1138	1140	95	3.83	-	3.11	-
16	Cuminal	16.984	1230	1224	95	-	3.20	-	-
17	Phellandral	17.995	1175	1252	93	-	1.42	3.39	-
18	p-Cymen-7-ol	18.340	1284	1274	91	-	1.40	6.23	-
19	p-Cymen-2-ol	18.524	1262	1200	92	-	-	1.19	7.02
20	Dodecamethylcyclohexasiloxane	18.693	-	-	-	-	-	-	23.2
21	Allo-aromandendrene	22.360	1386	1461	95	1.54	-	-	-
22	(E)-β-caryophyllene	22.904	1494	1467	94	0.81	-	-	-
23	Tetradecamethylcycloheptasiloxane	23.024	-	-	-	-	-	-	6.1
24	β-Nerolidol	25.368	1564	1565	97	1.78	-	-	-
25	Globulol	26.244	1530	1576	92	5.9	-	-	-
26	Spathulenol	26.501	1570	1619	94	1.59	-	2.26	-
27	Rosifoliol	27.349	1598	1600	87	0.85	-	-	-
28	Hexadecamethylcyclooctasiloxane	27.495	-	-	-	-	-	-	3.37
29	Cryptomeridiol	28.205	1738	-	88	0.65	-	-	-
**Total identified (%)**	-	-				99.84	99.4	98.73	99.1

* Experimentally determined Kováts retention indices; CEO: crude essential oil; F1: fraction 1; F2: fraction 2; F3: fraction 3.

**Table 2 pharmaceuticals-17-01552-t002:** Antioxidant activity of essential oil (EO) and fractions (F1, F2, F3) of *Eucalyptus globulus* compared to standards.

EO, Fractions, and Standards	DPPH IC_50_ (μg/mL)	ABTS IC_50_ (μg/mL)	FRAP EC_50_ (μg/mL)
**CEO**	ND	ND	22,402.20 ± 64.58 ^f^
F1	ND	ND	1054.93 ± 8.95 ^g^
F2	ND	9979.31 ± 122.41 ^c^	2103.51 ± 2.96 ^h^
F3	3329.34 ± 54.68 ^a^	3721.91 ± 27.02 ^d^	1185.48 ± 6.29 ^g^
Ascorbic acid	-	2.52 ± 0.02 ^e^	-
Quercetin	5.49 ± 0.02 ^b^	-	-
Catechin	-	-	13.90 ± 0.03 ^i^

The presented data are expressed as the mean ± standard deviation from three independent experiments. Significant differences are indicated by distinct superscript letters within the same column (*p* < 0.05). “ND” denotes “not determined”. IC_50_ refers to the concentration required to inhibit 50% of activity, while EC_50_ indicates the effective concentration that reduces 50% of Fe^3+^ to Fe^2+^.

**Table 3 pharmaceuticals-17-01552-t003:** Inhibition zones of essential oil from *Eucalyptus globulus* leaves, its fractions, and antibiotics against various strains of microorganisms.

Microorganisms	Inhibition Zone Diameter (mm) ^a^						
	Essential Oil				Antibiotics		
	CEO	F1	F2	F3	Ampicillin	Penicillin	Gentamicin
**Gram-positive**							
*Staphylococcus aureus*	14 ± 0.00 ^a^	13 ± 0.00 ^a^	46 ± 0.7 ^b^	45 ± 0.00 ^b^	07 ± 0.35	NA	22 ± 0.00
*Staphylococcus epidermidis*	11 ± 1.41 ^b^	13 ± 0.7 ^a^	08.00 ± 0.00 ^c^	10 ± 0.35 ^b^	NA	NA	14.33 ± 0.35
**Gram-negative**							
*Klebsiella pneumonia*	11 ± 0.7 ^a^	08.00 ± 0.00 ^b^	09 ± 0.35 ^b^	08 ± 0.00 ^b^	NA	NA	08 ± 0.7
*Enterobacter cloacae*	9.50 ± 0.35 ^a^	9.50 ± 0.35 ^a^	10.50 ± 0.35 ^a^	10 ± 0.35 ^a^	NA	NA	19.50 ± 0.35
*Escherichia coli*	9.00 ± 0.7 ^a^	09.50 ± 0.35 ^a^	11 ± 0.00 ^a^	08 ± 0.00 ^b^	NA	NA	20 ± 0.00
*Acinetobacter baumannii*	13 ± 0.00 ^b^	12.50 ± 0.00 ^b^	12 ± 0.7 ^a^	10 ± 0.00 ^c^	NA	NA	13 ± 0.7

Different letters on the same line are considered to be statistically significant when *p* < 0.05. Note: P5: penicillin (5 μg); GN 500: gentamicin (500 μg); AMP 10: ampicillin (10 μg). The diameter of the inhibition zone, which includes a disc diameter of 6 mm, was measured using the agar disc diffusion method. Each disc contained 15 μL of essential oil, along with 5 μg of penicillin, 500 μg of gentamicin, and 10 μg of ampicillin. NA: non-active; CEO: crude essential oil; F1: fraction 1; F2: fraction 2; F3: fraction 3.

**Table 5 pharmaceuticals-17-01552-t005:** Binding energy of volatile oil constituents complexed with the enzymes.

	PRDX5 (1H2D) Kcal/mol	*E. coli* (3G7B) Kcal/mol	*S. aureus* (6F86) Kcal/mol	*A. baumannii* (3TD3) Kcal/mol
Eucalyptol	−4.3	−4.6	−4.7	−4.8
α-Pinene	−4.3	−4.7	−4.6	−4.7
Cryptone	−4.4	−5.0	−5.4	−4.5
o-Cymene	−4.4	−5.2	−5.2	−4.4
p-Cymen-2-ol	−4.2	−5.3	−5.8	−4.4
Terpinen-4-ol	−4.4	−5.1	−5.6	−4.7

**Table 4 pharmaceuticals-17-01552-t004:** Minimum inhibitory concentration (MIC) and minimum bactericidal concentration (MBC) of essential oil from leaves of *Eucalyptus globulus* against various strains of microorganisms.

Microorganisms	MIC				MBC			
					Essential Oil			
	CEO	F1	F2	F3	CEO	F1	F2	F3
**Gram-positive**								
*Staphylococcus aureus*	10	2	2	1	10	2	2	10
*Staphylococcus epidermidis*	10	<10	10	4	10	-	10	10
**Gram-negative**								
*Klebsiella pneumonia*	<10	<10	<10	10	-	-	-	10
*Enterobacter cloacae*	<10	<10	10	10	-	-	10	10
*Escherichia coli*	<10	<10	<10	10	-	-	-	10
*Acinetobacter baumannii*	<10	<10	10	4	-	-	10	10

Note: CEO: crude essential oil. F1: fraction 1. F2: fraction 2. F3: fraction 3.

## Data Availability

Data are contained within the article.

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
