# Peer review of "GC–MS Characterization and Bioactivity Study of Eucalyptus globulus Labill. (Myrtaceae) Essential Oils and Their Fractions: Antibacterial and Antioxidant Properties and Molecular Docking Modeling"

_pharmaceuticals, 2024, doi:10.3390/ph17111552_

Round 1
Reviewer 1 Report
Comments and Suggestions for Authors
The objective of this paper is to characterize the essential oil of Eucaliptus. The added value of the manuscript is that the fractions of the essential oil is characterized instead of the overall essential oil. The essential oil is characterized by GC-MS to know the chemical composition and also in terms of antibacterial and antioxidant properties.
Comments:
The abstract should be rewritten. In my opinion, it should not contain “Background”, “Methods”, “Results” and “Conclusions”. All the ideas should be given as a whole in the abstract without specifying each section.
Line 102. Please correct “by(Ait-…). Please correct also the sentence: “However, other authors Hafsa et al. …”
Lines 108-109. Please rewrite the sentence: “These compounds, however, are completely absent from…”. In my opinion, the compounds can be in the essential oil but maybe the LOD of the developed GC-MS is not enough to detect these compounds previously found in this kind of essential oil. Instead of completely absent, it should be written that there were not detected by GC-MS. And that they could be present in the sample at lower concentrations than LOD.
Line 110. Please consider correcting “The diversity” by “The differences in the chemical composition can be explained by…”
Line 119. Please correct “respectfully” by “respectively”
For the GC-MS analysis it should be added:
- The uncertainty of the relative composition. The essential oil is analysed only once? Or some replicates are done? Usually, an uncertainty interval is given (at least based on the standard deviation of three replicates)
- The % of similarity should be given for each compound (related with the database of the MS spectra). The retention index is usually given in GC to confirm identification.
- Chromatograms of the fractions could be given (at least as supporting information). Also the mass spectra could be given.
- The relative composition is obtained as a percentage of areas. Is this area based on the whole MS spectrum of each component? This could be explained. Because in GC-MS quantification is usually done by SIM with a “quantifier ion” and here I guess the chromatograms are processed in terms of TIC (Total Ion Chromatogram)
Line 135. Please consider correct “et EC50…”. I guess the term in French “et” should be corrected by “and”.
The authors could explain how the measurements IC50 and EC50 were calculated. And how the uncertainty related to these terms is calculated. Usually the experimental plots are shown related to these kind of experiments. Maybe this could be shown at least in supporting information.
In my opinion, the section Material and Methods should be placed after the introduction and before the results. Please, pay attention in this section with the numeration. I would say there is an error with the number of “4.5. In vitro antimicrobial activity”. And the same with section 4.5.1.
Line 584. Please consider correcting the sentence “Integrating EO… requires precise analysis of their composition”. In analytical chemistry precise is not enough. It means that the result is very similar when the analysis is done several times but it does not mean that the results are correct. This should be corrected by “accurate”.
Author Response
Comments 1: The abstract should be rewritten. In my opinion, it should not contain “Background”, “Methods”, “Results” and “Conclusions”. All the ideas should be given as a whole in the abstract without specifying each section. Response 1: We have adhered to the journal's guidelines; however, I appreciate your suggestion. (page 1, Abstract, and line 31-48)
Comments 2: Line 102. Please correct “by (Ait-…). Please correct also the sentence: “However, other authors Hafsa et al. …” Response 2: Thank you for pointing this out. I agree with this comment; The term has been corrected. (page 1, Results and Discussion, and line 246-248)
|
Comments 3: Lines 108-109. Please rewrite the sentence: “These compounds, however, are completely absent from…”. In my opinion, the compounds can be in the essential oil but maybe the LOD of the developed GC-MS is not enough to detect these compounds previously found in this kind of essential oil. Instead of completely absent, it should be written that there were not detected by GC-MS. And that they could be present in the sample at lower concentrations than LOD.
Response 3: Thank you for pointing this out. I agree with this comment; The term has been corrected. (While our analysis did not detect these compounds in the essential oil, they may still be present at concentrations below the detection limit of the GC-MS employed. This aligns with findings from previous studies on similar essential oils.) (page 6, Results and Discussion, line 263-264)
Comments 4: Line 110. Please consider correcting “The diversity” by “The differences in the chemical composition can be explained by…”
Response 4: Thank you for pointing this out. I agree with this comment; The term has been corrected. (page 6, Results and Discussion, and line 255)
Comments 5: Line 119. Please correct “respectfully” by “respectively”
Response 5: Thank you for pointing this out. The term has been corrected. (page 6, Results and Discussion, line 262)
Comments 6: The uncertainty of the relative composition. The essential oil is analysed only once? Or some replicates are done? Usually, an uncertainty interval is given (at least based on the standard deviation of three replicates)
Response 6: Thank you for pointing this out. I agree with this comment; But in this present study, the essential oil was analyzed only once to identify its chemical composition and estimate the percentage of each compound. We recognize that presenting an uncertainty interval based on multiple replicates would enhance the robustness of our findings. The presentation of percentage ranges for each compound may be addressed in future research.
Comments 7: The % of similarity should be given for each compound (related with the database of the MS spectra). The retention index is usually given in GC to confirm identification.
Response 7: Thank you for pointing this out. Added in the table. (page 8, Results and Discussion)
Comments 8: Chromatograms of the fractions could be given (at least as supporting information). Also the mass spectra could be given.
Response 8: Thank you for pointing this out. Added in the manuscript. (page 7, Results and Discussion)
For the mass spectra Added in supporting information.
Comments 9: The relative composition is obtained as a percentage of areas. Is this area based on the whole MS spectrum of each component? This could be explained. Because in GC-MS quantification is usually done by SIM with a “quantifier ion” and here I guess the chromatograms are processed in terms of TIC (Total Ion Chromatogram)
Response 9: Thank you for your insightful question. In our analysis, the relative composition was indeed based on the total ion current (TIC) of each component in the chromatograms. We did not employ selective ion monitoring (SIM) with quantifier ions for this study.
Comments 10: Line 135. Please consider correct “et EC50…”. I guess the term in French “et” should be corrected by “and”.
Response 10: Thank you for pointing this out. The term has been corrected. (page 9, Results and Discussion, line. 286)
Comments 11: The authors could explain how the measurements IC50 and EC50 were calculated. And how the uncertainty related to these terms is calculated. Usually the experimental plots are shown related to these kind of experiments. Maybe this could be shown at least in supporting information.
Response 11: The results were expressed as the 50% inhibitory concentration (IC50) for DPPH and ABTS, calculated using the regression equation derived from plotting concentration against percentage inhibition. Similarly, the effective concentration (EC50) that reduces 50% of the Fe³⁺ ion to Fe²⁺ was determined by plotting concentration as a function of absorbance and employing the regression equation. We will consider including the relevant experimental plots in the supporting information to enhance clarity and transparency.
Comments 12: In my opinion, the section Material and Methods should be placed after the introduction and before the results. Please, pay attention in this section with the numeration. I would say there is an error with the number of “4.5. In vitro antimicrobial activity”. And the same with section 4.5.1.
Response 12: We have adhered to the journal's guidelines; however, I appreciate your suggestion and have moved the "Materials and Methods" section to follow the introduction. Additionally, I have corrected the numbering errors in sections 4.5 and 4.5.1, and the hierarchy of sections has been adjusted to ensure clarity. (page 3, Material and Methods, line. 101)

Reviewer 2 Report
Comments and Suggestions for Authors
The manuscript describes essential oil extraction from Eucalyptus globulus leaves by hydrodistillation, yielding different fractions depending on the extraction temperature. In addition, antioxidant and antimicrobial activity assessment data are provided. The manuscript shows a comprehensive approach to the experiment, however, the novelty in understanding the objectives of the experiment is clearly omitted. The authors should pay attention to the design of the article, especially in the Materials and Methods section. There is no statistical processing of the GC-MS data and antimicrobial samples. And the question of fractionation remains open. If the authors propose such an approach for the first time (as written in line 87), changing only one parameter (temperature during waterproofing) is not enough. Below are more detailed comments:
-the title is overloaded, should be optimized;
-pay attention to the requirements for the abstract.
If the authors emphasize the novelty in the fractionation of eucalyptus oil, the introduction should describe the previously used methods and their shortcomings.
It is suggested to carry out fractionation of essential oil only by the temperature parameter. For comparative analysis, more parameters should be used, duration, solvent, method (water distillation, steam distillation, water-steam and distillation), etc.
Line 87 - "However, it seems that these EOs have never been fractionated before" - this is not entirely true, because there are publications on the comparison of eucalyptus extraction methods.
at the beginning of the Results and Discussion section, a brief description of the fractions should be written, because it is currently unclear what they are and where they came from.
There is no statistical processing of the data.
More detailed information should be given about the place of collection of raw materials, identification and preservation of test samples in the herbarium.
The fractionation conditions should also be described in more detail.
The conclusion should be improved, but it is clear what is the advantage of fractionating oil over the usual method of obtaining. In the discussion, more attention should be paid to comparing your data and previously published data of other authors, taking into account the countries of origin of the plant, the conditions for obtaining oil.
Author Response
Comments 1: the title is overloaded, should be optimized. Response 1: Thank you for pointing this out. I agree with this comment; (GC-MS characterization and bioactivity study of Eucalyptus globulus Labill. (Myrtaceae) essential oils and their fractions: antibacterial and antioxidant properties and molecular docking modeling.)
Comments 2: pay attention to the requirements for the abstract. Response 2: Thank you for pointing this out. Revised in manuscript (page 1, Abstract, and line 31-48)
|
Comments 3: : If the authors emphasize the novelty in the fractionation of eucalyptus oil, the introduction should describe the previously used methods and their shortcomings.
Response 3: Thank you for pointing this out. I agree with this comment; Added in manuscript. (page 2, Introduction, line 89-92)
Comments 4: It is suggested to carry out fractionation of essential oil only by the temperature parameter. For comparative analysis, more parameters should be used, duration, solvent, method (water distillation, steam distillation, water-steam and distillation), etc.
Response 5: Thank you for pointing this out. I agree with this comment. In this study we initially extracted the essential oil of E. globulus by hydrodistillation, towards the end we obtained a crude essential oil, this oil was put into the fractionation apparatus, where separation was carried out by increasing the temperature of the same essential oil giving separate fractions.
Comments 5: Line 87 - "However, it seems that these EOs have never been fractionated before" this is not entirely true, because there are publications on the comparison of eucalyptus extraction methods.
Response 5: Thank you for pointing this out. I totally agree with you, raw essential oil extracted by different extraction methods (hydrodistillation, steam distillation...) and changing the parameters (time, technique...) is already consumed, but fractionation (dividing the same essential oil obtained after extraction into fractions by increasing the temperature of this oil) has not been studied before.
Comments 6: at the beginning of the Results and Discussion section, a brief description of the fractions should be written, because it is currently unclear what they are and where they came from.
Response 6: Thank you for pointing this out. I agree with this comment; Added in manuscript.
Comments 7: There is no statistical processing of the data.
Response 7: Thank you for pointing this out. Added in the manuscript. (page 12, Results and Discussion)
Comments 8: More detailed information should be given about the place of collection of raw materials, identification and preservation of test samples in the herbarium.
Response 8: Thank you for pointing this out. Thank you for pointing this out. The plant material was identified by Mohamed El Kadiri, botanist at the Faculté des Sciences de Tétouan, Université Abdelmalek Essaâdi, Morocco. The herbarium specimen has been deposited and given the reference number 1-130.881. (page 3, Material and Methods, line 105-107)
Comments 9: The fractionation conditions should also be described in more detail.
Response 9: Thank you for pointing this out. Precise fractionation conditions were established to maximize separation of components according to their volatility. A B-585 glass oven was used to ensure optimum thermal stability and minimize condensation losses. A 10 ml quantity of essential oil was introduced into a dedicated glass ball, enabling fractionation to be carried out in several controlled stages. The first fraction (F1) was collected at a temperature of 160°C, maintained for 10 minutes, in order to vaporize the most volatile compounds. Next, the temperature was raised to 180°C to obtain the second fraction (F2), also stabilized for 10 minutes, enabling the extraction of intermediately volatile compounds. Finally, a third residual fraction (F3) was taken from the initial ball after fractionation, containing components not volatilized at the previous temperatures. This progressive fractional distillation protocol was designed to ensure efficient separation of the various constituents, while limiting thermal degradation of sensitive compounds.
Comments 10: The conclusion should be improved, but it is clear what is the advantage of fractionating oil over the usual method of obtaining. In the discussion, more attention should be paid to comparing your data and previously published data of other authors, taking into account the countries of origin of the plant, the conditions for obtaining oil.
Response 10: Thank you for pointing this out. This question has been considered

Reviewer 3 Report
Comments and Suggestions for Authors
The authors have fractionated the volatile oil of Moroccan Eucalyptus globulus, studied the chemical composition of the resultant fractions by GC/MS. Then they investigated their antioxidant activity using DPPH, Frap and ABTs assays. Furthermore, the authors evaluated the antimicrobial potential of the crude volatile oil and its fractions against some gram +ve and gram -ve bacteria. The authors used docking studies to support their experimental findings. The manuscript is interesting but it needs language revision (there are numerous typos). Moreover, there are some concerns should be addressed
1. In the material and methods, the names and the source of the tested microorganisms should be provided.
2. The methodology of the MBC determination is missing.
3. For better representation of the results, photos of the inhibition zones should be added.
4. The antioxidant activities of Eucalyptus globulus volatile oil and its fraction are rather low
5. The chromatograms of the GC analysis should be provided.
6. Regarding the docking studies, please rewrite that part as it is a bit disorganized. The authors should justify their choice of these targets; peroxiredoxin 5 enzyme, DNA gyrase from E. coli and S. aureus. Why the authors investigated the binding affinity of the compounds towards the DNA gyrase from E. coli (PDB ID: 3G7B) despite the tested sample showed better activity on Acinetobacter baumannii therefore, a molecular target linked to this microorganism should investigated instead. Moreover, binding affinities of other major compounds in the volatile oil and fractions were not studied.
Comments on the Quality of English LanguageThe manuscript need extensive language revision
Author Response
Comments 1: In the material and methods, the names and the source of the tested microorganisms should be provided. Response 1: Thank you for pointing this out, Added in manuscript. (page 4, Material and Methods, and line 163-168)
Comments 2: The methodology of the MBC determination is missing. Response 2: Thank you for pointing this out, Added in manuscript. (page 5, Material and Methods, and line 200-206)
|
Comments 3: For better representation of the results, photos of the inhibition zones should be added.
Response 3: Thank you for pointing this out. I agree with you about adding images, but I would prefer them to be in the supplementary information.
Comments 4: The antioxidant activities of Eucalyptus globulus volatile oil and its fraction are rather low
Response 4: Thank you for pointing this out. I agree with your observation concerning the relatively low antioxidant activities of Eucalyptus globulus volatile oil and its fractions. This finding may indeed depend on the specific protocol we used. However, according to the literature, I have found that similar studies report even lower antioxidant results compared to our results.
Comments 5: The chromatograms of the GC analysis should be provided.
Response 5: Thank you for pointing this out. Added in the manuscript. (page 7, Results and Discussion)
For the mass spectra Added in supporting information.
Comments 6: Regarding the docking studies, please rewrite that part as it is a bit disorganized. The authors should justify their choice of these targets; peroxiredoxin 5 enzyme, DNA gyrase from E. coli and S. aureus. Why the authors investigated the binding affinity of the compounds towards the DNA gyrase from E. coli (PDB ID: 3G7B) despite the tested sample showed better activity on Acinetobacter baumannii therefore, a molecular target linked to this microorganism should investigated instead. Moreover, binding affinities of other major compounds in the volatile oil and fractions were not studied.
Response 6: Thank you for pointing this out. I agree with this comment;
- Regarding the docking studies, please rewrite that part as it is a bit disorganized.
- Response: the docking results part has been rewritten.
- The authors should justify their choice of these targets; peroxiredoxin 5 enzyme, DNA gyrase from E. coli and S. aureus.
- Response: the choice of the target : peroxiredoxin 5 enzyme was based on the fact that, the in vitro test were not made on actual enzymes and we had to chose a target based on prior published articles. And it was chosen for its essential role in neutralizing reactive oxygen species. DNA gyrase from E. coli and S. aureus were chosen as docking target based as well on various published articles and without a prior knowledge on what are the in vitro microorganism tested are. The docking study was done separately.
- Why the authors investigated the binding affinity of the compounds towards the DNA gyrase from E. coli (PDB ID: 3G7B) despite the tested sample showed better activity on Acinetobacter baumannii therefore, a molecular target linked to this microorganism should investigated instead.
- Respond: a docking study on Acinetobacter baumannii was added.
- Moreover, binding affinities of other major compounds in the volatile oil and fractions were not studied.
- we have focused our study mainly on the most active volatile oil compounds, identified in previous research.
3. Response to Comments on the Quality of English Language |
|
Response 1: Thank you for pointing this out. English has been revised |

Round 2
Reviewer 1 Report
Comments and Suggestions for Authors
The authors have modified the manuscript according to almost all my comments. It is suitable for publication.
Just two minor comments. They could consider deleting lines 710-711 from the template and give the authors' contribution directly. There is also a supporting information with a title in French. You might consider deleting the French title.
Author Response
Comments: Just two minor comments. They could consider deleting lines 710-711 from the template and give the authors' contribution directly. There is also a supporting information with a title in French. You might consider deleting the French title.
Response : Thank you for your comments. After verification, we have corrected the errors mentioned. Lines 710-711 have been adjusted to show authors' contributions directly, and the French title has been removed.

Reviewer 2 Report
Comments and Suggestions for Authors
how the % content of components in Table 1 was calculated and how many repetitions were there for each sample.
Author Response
Comment: how the % content of components in Table 1 was calculated and how many repetitions were there for each sample.
Response : Thank you for pointing this out. Peak areas are calculated automatically after integration by the software.
The percentage of component content was calculated by dividing the area of each peak by the sum of the areas of all peaks, then multiplying by 100.
As far as the number of repetitions is concerned, a single measurement was carried out for each sample.

Reviewer 3 Report
Comments and Suggestions for Authors
Thank you for your favorable response to my comments. The only concern is about the tested microorganisms how they were identified if they are not reference strains (ATCC for example), are they clinical isolates? if so please provide the ethical code and way of identification ( you cam cite a reference .
Author Response
Comment: The only concern is about the tested microorganisms how they were identified if they are not reference strains (ATCC for example), are they clinical isolates? if so please provide the ethical code and way of identification ( you cam cite a reference).
Response : Thank you for your comment, The strains used in this study were supplied by the microbiology department of the medical analysis laboratory. They were derived from clinical isolates collected as part of the laboratory's regular diagnostic activities. No specific ethical approval code is associated, but the samples were processed in accordance with standard laboratory procedures and current institutional regulations.
Round 3
Reviewer 2 Report
Comments and Suggestions for Authors
The authors correct the comments. The manuscript may be accepted for publication.